# Porcine Teschovirus 2 3C^pro^ Evades Host Antiviral Innate Immunity by Inhibiting the IFN-β Signaling Pathway

**DOI:** 10.3390/microorganisms13061209

**Published:** 2025-05-26

**Authors:** Xin-Yu Zhang, Yu-Ying Li, Yi-Min Zhou, Wei Chen, Lu-Lu Xie, Yan-Qing Hu, Yan Qin, Hai-Xin Huang, Lin Zhou, Tian Lan, Wen-Chao Sun

**Affiliations:** 1Wenzhou Key Laboratory for Virology and Immunology, Institute of Virology, Wenzhou University, Wenzhou 325035, China; 18011456830@163.com (X.-Y.Z.); 24451041042@stu.wzu.edu.cn (Y.-M.Z.); 23451039005@stu.wzu.edu.cn (W.C.); 24451041033@stu.wzu.edu.cn (L.-L.X.); huyanqing0811@163.com (Y.-Q.H.); qinyan_qy@163.com (Y.Q.); huanghaixinn@163.com (H.-X.H.); q103937513@outlook.com (L.Z.); 20200493@wzu.edu.cn (T.L.); 2College of Animal Sciences, Institute of Preventive Veterinary Medicine, Zhejiang University, Hangzhou 310000, China; 15858559633@163.com; 3College of Veterinary Medicine, Northwest A&F University, Yangling 712100, China

**Keywords:** porcine teschovirus, 3C^pro^, NF-κB, inhibitor, protease activity

## Abstract

Porcine teschovirus (PTV) circulates in pig populations, causing clinical diseases such as poliomyelitis, reproductive disorders, and pneumonia. However, the molecular mechanisms underlying the pathogenesis of PTV infection have not been fully elucidated. Here, we found that PTV infection does not activate the promoters of NF-κB or IFN-β. The expression of PTV 3C^pro^ inhibits the promoter activity of NF-κB and IFN-β stimulated by SeV and inhibits the downstream transcription of NF-κB and IFN-β by blocking the phosphorylation and nuclear translocation of NF-κB. Coimmunoprecipiation (co-IP) experiments demonstrated that 3C^pro^ and NF-κB interact. The degradation of NF-κB was unaffected by inhibitors targeting lysosomes (NH4Cl), proteasomes (MG132), or caspases (Z-VAD-FMK). The protease activity of 3C^pro^, which relies on its catalytic active site, is vital for NF-κB cleavage and degradation. Loss of proteolytic activity in mutants abolished NF-κB degradation, impairing the ability of 3C^pro^ to suppress SeV-induced innate immunity and restore VSV-GFP replication, thereby underscoring its critical role in immune evasion by targeting NF-κB. This study reveals novel mechanisms underlying PTV-mediated suppression of host innate immunity.

## 1. Introduction

Porcine teschovirus (PTV) infection can lead to polioencephalomyelitis, reproductive disorders in sows, enteric diseases, and pneumonia, constituting a substantial threat to swine production [1]. PTV is a single-stranded positive-sense RNA virus belonging to the *Teschovirus* genus in the *Picornaviridae* family. The virus particles are spherical, 20–30 nanometers in diameter, and lack an envelope [2]. The genome is approximately 7.2 kb long and has an open reading frame encoding a polyprotein that is cleaved into four structural proteins (VP1-VP4) and eight nonstructural proteins (L^pro^, 2A-2C, 3A-3D) [3]. In 2019, the International Committee on Taxonomy of Viruses (ICTV) proposed a revision of the teschovirus taxonomy; PTV currently comprises at least 15 PTV genotypes, which has expanded with the recent discoveries of novel genotypes [4]. In 1929, the first cases of infection with PTV were documented in Teschen, Czechoslovakia, which was later recognized as Teschen disease [5]. Pigs are the only hosts, with piglets being the most susceptible. The virus primarily resides in the swine encephalon and spinal cord, and infection can result in up to 100% mortality in piglets. The disease spreads mainly through subclinical infections, making early detection difficult [6]. In the past twenty years, PTV has spread progressively throughout Asia, including countries such as Japan [7], India [8], and China [9]. Moreover, outbreaks have been documented in North American regions, including Canada [10], the United States [11], and Haiti [12]. However, effective treatments are still lacking, highlighting the significance of investigating and preventing PTV infections.

A great research question is how PTV can evade innate immunity and spread through replication. Research has shown that *Picornaviruses* have evolved strategies to counteract the host’s innate immune system [13]. The picornavirus 3C protease functions as a chymotrypsin-like enzyme that cleaves viral proteins to form mature proteins, playing a crucial role in viral replication. It can also combat innate immunity by regulating host factors and promoting viral replication [14]. In addition, 3C^pro^ broadly suppresses type I interferon signaling, facilitating viral immune evasion [15]. Seneca Valley virus (SVA) 3C^pro^ induces caspase-3-dependent NF-κB-p65 proteolysis, disrupting survival signaling and triggering apoptotic cell death [16]. SVA 3C^pro^ degrades RIG-I, thereby inhibiting its ability to recognize viral RNA and suppress IFN-β-mediated innate immunity [17]. Enterovirus 71 (EV71) 3C^pro^ cleaves TIR-domain-containing adapter-inducing IFN-β (TRIF) at Q312, impacting NF-κB activation and IFN-β production [18]. Encephalomyocarditis virus (EMCV) 3C^pro^ cleaves TANK and disrupts the TANK–TBK1–IKKε–IRF3 complex, which results in a reduction in IRF3 phosphorylation and IFN-β production [19].

Previous studies have shown that PTV VP1 targets melanoma differentiation-associated gene 5 (MDA5) through its 2CARD and Hel domains to effectively inhibit IFN-β production and block NF-κB activation [20]. PTV2 infection can induce incomplete autophagy, which impedes viral propagation within PK-15 cells [21]. Although significant progress has been made in understanding the pathogenesis and immune evasion strategies of PTV, further investigation is still needed. Given the crucial role of picornaviral 3C^pro^ in viral replication and host antiviral pathway suppression, elucidating its functional mechanisms in PTV immune evasion holds important theoretical significance and research value. Therefore, this study aimed to elucidate the functional significance of PTV 3C^pro^ in interfering with key antiviral pathways. By exploring these mechanisms, we hope to provide new insights into PTV pathogenesis and contribute to the development of targeted interventions against this important swine pathogen.

## 2. Materials and Methods

### 2.1. Cells, Viral Strains, and Plasmids

ST cells (ATCC, CRL-1746, Manassas, VA, USA), HEK-293T cells (ATCC, CRL-11268), Dulbecco’s modified Eagle’s medium (Gibco, Carlsbad, CA, USA), and fetal bovine serum (Cellmax, SA101.02, Beijing, China) were used. PTV strain type 2 GX/2020, Sendai virus (SeV), and recombinant vesicular stomatitis virus expressing green fluorescent protein (VSV-GFP) were stored at the Wenzhou Key Laboratory for Virology and Immunology. The NF-κB plasmid (NCBI, NM_001114281.1) was constructed by Xin-yu Zhang, and the H-NF-κB, NF-κB-Luc, IFN-β-Luc, and PRL-TK plasmids were also maintained in the laboratory.

### 2.2. Antibodies

Anti-Flag (Bioss, bsm-33346 M, Beijing, China), anti-NF-κB p65 rabbit polyclonal (Beyotime, AF0246, Shanghai, China), anti-phospho-NF-κB p65 (Ser536) (93H1), anti-Myc (Bioss, bsm-51003 M), rabbit anti-Lamin B1 (Bioss, bs-34005R), anti-rabbit IgG (H + L), F(ab’)2 fragment (Cell Signaling Technology, 8889s, Shanghai, China), anti-mouse IgG (H + L), F(ab’)2 fragment (CST, 4408S, Shanghai, China) and GAPDH (Proteintech, 60004-1-Ig, Wuhan, China) antibodies were used.

### 2.3. Luciferase Reporter Assay

Briefly, 293T or ST cells were seeded in 12-well plates and cotransfected with reporter plasmids encoding NF-κB-Luc or IFN-β-Luc, along with TK-Renilla and other necessary expression plasmids, using ExFect Transfection Reagent (Vazyme, T101, Nanjing, China). An empty vector served as a negative control. A TransDetect^®^ Double-Luciferase Reporter Assay Kit (TransGen, Lot# R30906, Beijing, China) was used for dual luciferase detection. The luciferase reaction reagent and luciferase reaction reagent II were sequentially added according to the manufacturer’s instructions. The luminescence values of Luc and TK were measured using a microplate reader (BioTek, Synergy H1, Winooski, VT, USA), and the luciferase activity was normalized to firefly luciferase and Renilla luciferase values. All data from the reporter gene assays were derived from three independent experiments.

### 2.4. RNA Extraction and Plasmid Construction

Following the manufacturer’s protocol, RNA was isolated from cells using a total RNA extractor (Sangon, B511311, Shanghai, China). cDNA synthesis was subsequently performed with an RT–PCR system (TaKaRa, RR014A, Beijing, China). PCR amplification was conducted using the primer pairs detailed in Table 1. According to the instructions included with the high-fidelity PCR Master Mix (Vazyme, P525), the PCR protocol was set as follows: initial heat activation at 95 °C for 3 min, followed by 35 cycles of amplification (each cycle consisting of 95 °C denaturation for 15 s, 58 °C annealing for 15 s, and 72 °C extension for 1 min), with a final extension at 72 °C for 5 min.

The amplified product was purified using a gel extraction system (US Everbright, UE-GX-250, Suzhou, China). The purified DNA fragment was then cloned and inserted into the digested vector using a cloning system (Vazyme, C115). Following transformation into Fast T1 (Vazyme, C505) by heat shock treatment, positive transformants were selected using colony PCR after 12 h of incubation and subsequently verified by sequencing analysis (Sangon).

### 2.5. Real-Time PCR Assay

After reverse transcription of RNA into cDNA, quantitative PCR amplification was performed with a qPCR Master Mix (Vazyme, Q712) and specific primers (Table 2) on an ABI 7500 Fast Real-Time PCR instrument, with GAPDH used as the internal control. qPCR was carried out after preparing a reaction mixture consisting of 10.0 μL of qPCR Master Mix, 0.4 μL of each primer, and 1.0 μL of cDNA, with nuclease-free water added to achieve a final volume of 20.0 μL. qPCR amplification was carried out with an initial denaturation step at 95 °C for 30 s, followed by 40 cycles consisting of denaturation at 95 °C for 10 s and combined annealing/extension at 60 °C for 30 s. The 2^−ΔΔCT^ method was employed to calculate the relative mRNA expression levels.

### 2.6. Coimmunoprecipitation (Co-IP), Nuclear and Cytoplasmic Protein Extraction, and Western Blotting

Following 24 h of incubation after PTV 3C^pro^ transfection, the ST cells were subjected to 12 h of SeV infection. The cytoplasmic and nuclear proteins were separately extracted according to the instructions of the Nuclear and Cytoplasmic Protein Extraction Kit (Beyotime, P0027).

To perform the co-IP experiments, NP-40 lysis buffer containing PMSF was prepared. The cells were lysed on ice for 10 min. After centrifugation at 12,000 rpm for 10 min, the supernatant was collected and mixed with specific antibodies, followed by overnight incubation at 4 °C. At room temperature, a 60-min incubation was performed with the antibody-bound proteins and magnetic beads (ThermoScientific™ 88802, Waltham, MA, USA). The eluted beads were mixed with 1× SDS buffer, boiled, and used for western blot analysis.

Equal amounts of protein samples were separated using SDS–PAGE and transferred onto PVDF membranes. The membranes were then blocked with 5% skim milk at room temperature for 1 h and incubated with the corresponding primary antibodies overnight at 4 °C. After washing, the membranes were incubated for 60 min at room temperature with HRP-conjugated secondary antibodies, followed by chemiluminescent detection (ChemiDoc™ XRS+, Bio-Rad, Hercules, CA, USA).

Grayscale analysis was performed using ImageJ1.54m software. Images were converted to 8-bit grayscale mode, and background noise was removed using default parameters. After the image was inverted, the area, mean gray value, and integrated density were measured. The data were normalized for subsequent statistical analysis.

### 2.7. Gene Site-Directed Mutagenesis Experiment

Table 3 summarizes the oligonucleotides designed for introducing specific mutations. PCR was carried out on the 3C^pro^ plasmid using either the sense primer PTV-3C-F together with an antisense primer harboring the target mutation, or vice versa. This reaction yielded a DNA fragment encompassing the mutated region. The PCR amplification was performed using the same protocol as described in Section 2.3, except with an annealing temperature of 65 °C. The DNA fragment was extracted from the gel and subsequently utilized as a primer in PCR amplification to generate the complete mutated 3C^pro^ gene. After the mutated gene was connected to the vector, it was transformed into chemically competent FAST T1 cells, and plasmid preparation and identification were performed to obtain genes with site-specific mutations.. Single-gene site-directed mutagenesis experiments revealed that the 3C_DM_ double mutant plasmid was successfully constructed by repeating the aforementioned experimental procedures, such as introducing the C158 mutation into the H49A plasmid or the H49 mutation into the C158A plasmid.

### 2.8. Indirect Immunofluorescence Assay (IFA)

Following seeding in glass dishes, the ST cells were transfected with the NF-κB and 3C^pro^ plasmids and then exposed to SeV infection 24 h later, with samples collected at 12 hpi. Following fixation with 4% paraformaldehyde (30 min), the cells were permeabilized with 0.5% Triton X-100 (10 min) and blocked with 5% BSA (1 h). Primary antibody incubation was performed overnight at 4 °C, followed by incubation with secondary antibodies (1 h, RT). Nuclear staining with DAPI (5 min) preceded fluorescence microscopy imaging.

### 2.9. VSV Infection Experiment

Briefly, 293T cells in 12-well plates were transfected for 24 h with either the empty vector or the 3C^pro^ plasmid. Following medium replacement with 2% serum solution, VSV-GFP and poly(I:C) were introduced. After 12 h, the viral fluorescence inside the cells was observed under a fluorescence microscope and photographed. The cells were gently scraped off with a cell scraper and centrifuged at 4000 rpm for 5 min. The supernatant was discarded, and the cells were collected and gently suspended in PBS. Flow cytometry was performed, and the results were analyzed and processed using the professional software FlowJo10.10.

### 2.10. Statistics

All the experiments were performed in triplicate. The data from the bar charts were analyzed using GraphPad Prism 9 and are expressed as the means ± standard deviations (SDs). Statistical significance was determined at *p* < 0.05; * *p* < 0.05, ** *p* < 0.01, *** *p* < 0.001, and **** *p* < 0.0001; NS indicates no significant difference.

## 3. Results

### 3.1. PTV Infection Inhibits IFN-β Promoter Activation

Interferons (IFNs) are crucial cytokines with diverse antiviral and cellular regulatory functions. IFN-β plays a pivotal role in antiviral defense by triggering downstream pathways to stimulate an antiviral response. However, the mechanisms by which PTV evades innate immunity and the effects of IFN-β remain elusive. We evaluated whether PTV infection can induce the promoter activity of NF-κB and IFN-β. According to the dual-luciferase reporter assay, PTV infection failed to activate the NF-κB or IFN-β pathways (Figure 1A). Further studies revealed that PTV infection could inhibited the SeV-induced activity of the NF-κB and IFN-β promoters (Figure 1B). These findings suggest that PTV is capable of evading the host immune response.

### 3.2. PTV 3C^pro^ Decreases NF-κB and IFN-β mRNA Levels and Inhibits Promoter Activity

The 3C^pro^ protease in various *Picornaviridae* strains usually has an inhibitory effect on the IFN-β pathway. To investigate whether PTV 3C^pro^ similarly possesses this ability, we conducted a series of experimental analyses. A dual-luciferase reporter assay was used to evaluate the impact of 3C^pro^ on the activity of the NF-κB and IFN-β promoters, which revealed that PTV 3C^pro^ inhibited the NF-κB and IFN-β promoter activity induced by SeV stimulation (Figure 2A). In addition, the results of fluorescent quantitative PCR revealed that under SeV stimulation, the mRNA levels of NF-κB and IFN-β in 3C^pro^-transfected cells were lower than those in empty vector-transfected cells (Figure 2B). Furthermore, the effects of 3C^pro^ on NF-κB promoter activity induction by upstream signaling components was analyzed using a dual-luciferase reporter system. The NF-κB promoter activity induced by MDA5, RIG-I, MAVS, and TBK1 was similarly inhibited by PTV 3C^pro^ (Figure 2C). These data collectively establish PTV 3C^pro^ as a transcriptional suppressor of NF-κB, and we proceeded to evaluate whether it exerts its inhibitory effects by disrupting NF-κB phosphorylation and nuclear translocation.

### 3.3. PTV 3C^pro^ Inhibits the Phosphorylation and Translocation of NF-κB

Typically, NF-κB activation involves its phosphorylation and translocation to the nucleus and binding to specific promoter sequences to exert its effects. Through indirect immunofluorescence experiments, 3C^pro^ inhibited the nuclear translocation of NF-κB induced by SeV stimulation (Figure 3A). An examination of NF-κB levels in the cytoplasm and nucleus revealed a significant reduction in NF-κB levels in both compartments (Figure 3B). The phosphorylation of NF-κB at Ser536, triggered by diverse stimuli, facilitates its nuclear translocation and increases target gene expression. After PTV 3C^pro^ was transfected into ST cells, the protein levels of NF-κB and phosphorylated NF-κB were analyzed using western blotting at different time points after SeV stimulation. The results showed that PTV 3C^pro^ decreased the protein levels of NF-κB and phosphorylated NF-κB (Figure 3C). Multiple experiments have demonstrated that 3C^pro^ can reduce the level of phosphorylated NF-κB, thus preventing its nuclear translocation.

### 3.4. Mechanism of 3C^pro^-Mediated Degradation of NF-κB

In previous experiments, we reported that PTV 3C^pro^ decreases NF-κB protein levels. To verify the species specificity of PTV 3C^pro^ for NF-κB, PTV 3C^pro^ was transfected into 293T cells, where it was found to cleave the human NF-κB protein (H-NF-κB) (Figure 4A). Under the assumption that 3C^pro^ interacts with NF-κB or H-NF-κB, co-IP experiments revealed the interaction between NF-κB or H-NF-κB and 3C^pro^ (Figure 4B). The degradation and cleavage pathways were subsequently explored. The primary protein degradation pathways in eukaryotic cells include lysosomal, ubiquitin-proteasome, and caspase-mediated pathways. Cotransfection of 3C^pro^ and NF-κB (or H-NF-κB) was performed, and 24 h later, NH_4_Cl, MG132, and Z-VAD-FMK—which target lysosomes, proteasomes, and caspases, respectively—were added. The results revealed that NF-κB was inhibited in both species, but that the mechanisms of inhibition were different. NH_4_Cl, MG132, and Z-VAD-FMK did not block NF-κB degradation, whereas Z-VAD-FMK prevented H-NF-κB degradation but did not affect H-NF-κB cleavage (Figure 4C). Moreover, 3C^pro^ has serine/cysteine protease characteristics and features an active site comprising Cys-His residues. By comparing the amino acid sequences of PTV 3C^pro^ with those of other members of the *Picornaviridae* family, the catalytic site of PTV 3C^pro^ was identified (Figure 4D). Three mutant variants—H49A, C158A, and a double mutant (DM)—were subsequently generated for further investigation. The catalytic residues H49 and C158 were conserved across all the analyzed PTV genotypes (Appendix A). Next, PTV 3C^pro^ and its mutants were cotransfected with NF-κB (or H-NF-κB) into 293T cells. According to the western blotting results, the PTV 3C^pro^ mutant no longer cleaved or inhibited H-NF-κB nor NF-κB (Figure 4E). These findings suggest that the capacity of PTV 3C^pro^ to inhibit NF-κB depends on its protease activity.

### 3.5. The PTV 3C^pro^ Mutant Loses the Ability to Evade Innate Immunity

To further elucidate the immunomodulatory function of the 3C^pro^ mutant, the impact of the PTV 3C^pro^ mutant on IFN-β or NF-κB promoter activity was studied using a dual-luciferase reporter assay. The experimental findings revealed that site-directed mutagenesis at the enzymatic active site abrogated the ability of PTV 3C^pro^ to suppress both IFN-Luc and NF-κB-Luc promoter transcriptional activities (Figure 5A). Virus replication was assessed by measuring the fluorescence intensity of vesicular stomatitis virus (VSV-GFP). A significant reduction in GFP fluorescence intensity was observed in cells treated with poly(I:C), which activates NF-κB and IFN-β, suggesting that VSV-GFP replication was inhibited. However, GFP fluorescence intensity was recovered in cells transfected with 3C^pro^, suggesting a partial restoration of VSV-GFP replication, whereas no recovery was observed when the mutant was expressed. Fluorescent cells were quantified using flow cytometry, and the results also revealed that cells transfected with PTV 3C^pro^ had greater fluorescence intensity than cells transfected with the mutant (Figure 5B,C). These results demonstrate that 3C^pro^ can suppress the activation of innate immunity by poly(I:C), but the mutant cannot suppress NF-κB activation and inhibit the innate immune response, thereby losing the ability to evade innate immunity and aid in virus replication.

## 4. Discussion

PTV is widely distributed among swine populations worldwide and is commonly found in the feces of asymptomatic pigs, revealing its ability to establish intestinal colonization without inducing clinical symptoms [22,23]. However, highly pathogenic PTV strains can cause severe clinical symptoms, including moderate to severe neurological symptoms, respiratory diseases, and inflammatory responses [22]. The primary mode of PTV transmission is through oral ingestion, but exposure to urine and feces can also lead to infection. Following oral infection, PTV primarily replicates in the tonsils and intestines [1]. Characteristic signs after 1–2 d of PTV infection include elevated body temperature, diarrhea, and flaccid and spasmodic paralysis lasting for approximately 10–11 d; respiratory failure represents the primary factor leading to mortality in infected hosts [24]. Moreover, the humoral immune response mediated by IgG and IgM antibodies plays a major role in combating PTV infections; locally produced IgA antibodies have been shown to have a protective effect when the virus enters through the oral cavity. The primary antibody class involved in fetal responses to the PTV is IgM, with IgG appearing as the next most common type [25,26].

In China, swine-related diseases caused by PTV were first reported in 2003 with the isolation of swine/CH/IMH/03 [27]. Since then, PTV strains have been sporadically discovered in several provinces in China, and infected swine typically had mild symptoms. Historically, high mortality from PTV infections has been associated mainly with the PTV-1 strain [12,28]. In 2021, a PTV-1 outbreak was reported in Northeast China, with an approximately 10% mortality rate among piglets [27,29]. Additionally, a significant outbreak of PTV-2 was documented by Wenqi Liang in 2023 at a pig farm located in Henan Province, China [30]. The emergence of the HeNZ1 strain resulted in severe neurological symptoms and diarrhea among piglets, along with very high morbidity and mortality rates among suckling pigs, marking the first reported serious outbreak caused by the PTV-2 strain [30]. Research on how PTV-2 evades the innate immune system response can help gain insights into its pathogenic mechanisms, leading to the development of effective treatment strategies to combat potential future outbreaks. *Picornaviridae* 3C^pro^ is a protease-like protein that contains a conserved His-Asp-Cys catalytic triad. This triad can cleave viral proteins to form mature proteins and promote viral replication by antagonizing host factors and innate immunity [15]. Although PTV has been causing substantial economic impacts on swine production for decades, investigations into its pathogenic mechanisms, particularly how it evades host innate immune responses, remain rare. In light of this gap, the present study focused on the potential immunomodulatory function of PTV 3C^pro^ in suppressing antiviral defense pathways.

We found that PTV infection does not increase the mRNA levels or activate the promoters of NF-κB and IFN-β in cells. Additionally, 3C^pro^ inhibits the promoter activity of NF-κB induced by upstream factors. Normally, phosphorylation of NF-κB at Ser536 leads to its nuclear translocation and binding to specific promoter sequences, activating the transcription of target genes [31]. This activation leads to the formation of a complex and the initiation of a partially overlapping transcriptional program, which in turn suppresses viral replication [32,33]. The pivotal role of the antiviral innate immune response in protecting cells from viral infections has been extensively demonstrated. For example, IFN-β is essential for the regulation of virus replication and has been shown to effectively reduce the viral load in cardiomyocytes infected with coxsackievirus B3 [34,35]. However, our experiments revealed that 3C^pro^ inhibits the phosphorylation and nuclear translocation of NF-κB through degradation. By studying the degradation mechanism of NF-κB, we found that PTV 3C^pro^ has different inhibitory effects on NF-κB in different species, such as cleaving human NF-κB and degrading porcine NF-κB. While Z-VAD-FMK, a pancaspase inhibitor, suppresses the degradation of H-NF-κB, it does not block its cleavage. The degradation of porcine NF-κB is unaffected by the lysosome inhibitor NH_4_CL, the proteasome inhibitor MG132, and the caspase inhibitor Z-VAD-FMK. Co-IP experiments revealed that 3C^pro^ interacts with both forms of NF-κB. The mechanism of PTV 3C^pro^-mediated caspase 3 cleavage of NF-κB is species specific, suggesting that human NF-κB has a cleavage site that is recognized by 3C^pro^, whereas the sequence of porcine NF-κB differs at this site.

Previous studies have highlighted the essential role of proteolytic activity in maintaining the functional integrity of the 3C protein. For example, inactivation of PSV 3C^pro^ prevents MAVS cleavage and abolishes its ability to degrade TBK1 and MDA5 [36]. Similarly, SVA 3C^pro^ loses both cleavage and degradation activities toward RIG-I and MAVS upon inactivation [21]. To explore this mechanism in PTV, we performed multiple sequence alignments between PTV 3C^pro^ and other members of the *Picornaviridae* family to identify conserved catalytic residues. On the basis of these findings, we generated site-directed mutants lacking protease activity. Functional assays revealed that inactivated PTV 3C^pro^ could no longer cleave or induce the degradation of NF-κB. Moreover, VSV-GFP experiments demonstrated that the mutant protein failed to suppress IFN-β signaling activation. Dual-luciferase assays revealed that the PTV 3C^pro^ mutant no longer inhibited the activity of the IFN-β promoter. Future research should validate these findings in in vivo porcine models to assess how 3C^pro^-mediated immune suppression impacts viral pathogenesis and tissue tropism. Comparative analyses of PTV-1 and PTV-2 strains, including the HeNZ1 variant, are critical for determining whether genotype-specific 3C^pro^ activity is correlated with virulence and immune evasion efficiency. Reverse genetic approaches should generate 3C^pro^-mutant PTV viruses to evaluate their impact on immune evasion; however, prior to assessing immune evasion, the effect of 3C^pro^ inactivation on viral replication must be established.

In summary, our findings demonstrate that PTV 3C^pro^ plays a critical role in evading the host innate immune response by binding with NF-κB, inhibiting its phosphorylation and nuclear translocation. Notably, we show that PTV 3C^pro^-induced degradation of NF-κB is species specific. The results of mutagenesis studies further confirmed that the protease activity of 3C^pro^ is essential for these immunosuppressive functions. These results provide novel insights into the immune evasion strategies of PTV and highlight the importance of 3C^pro^ as a potential target for antiviral intervention.

## Figures and Tables

**Figure 1 microorganisms-13-01209-f001:**
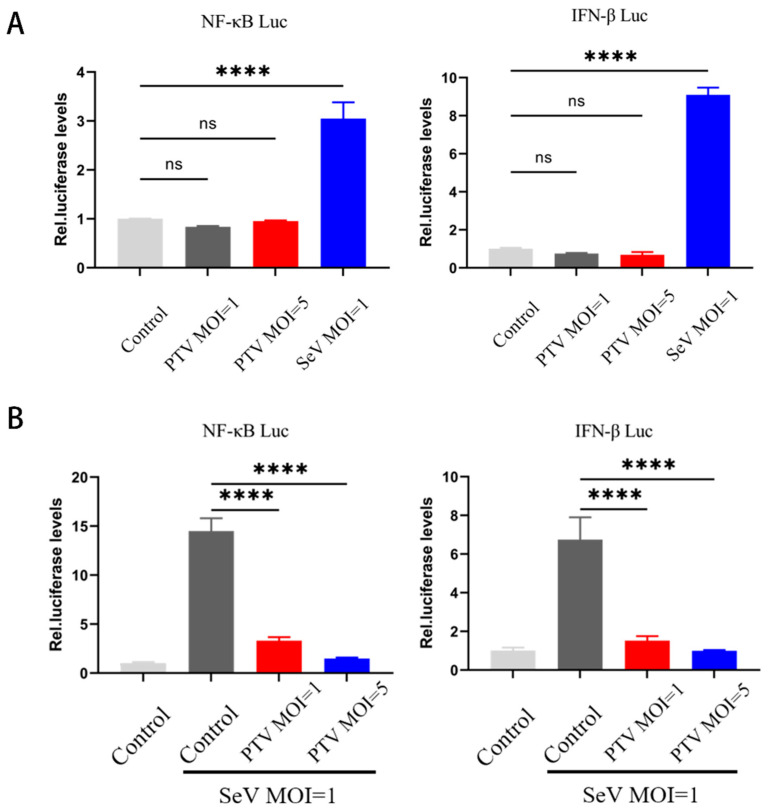
PTV infection inhibits the activity of the NF-κB and IFN-β promoters. (**A**) ST cells were cotransfected with NF-κB-Luc or IFN-β-Luc (1000 ng) and PRL-TK (50 ng), followed by 24 h of infection with PTV (MOI = 1, 5) or SeV (MOI = 1), and the fluorescence intensity was measured. (**B**) ST cells were cotransfected with NF-κB-Luc or IFN-β-Luc (1000 ng) and PRL-TK (50 ng), followed by 24 h of induction with SeV (MOI = 1) and 24 h of infection with PTV (MOI = 1, 5), and the fluorescence intensity was measured. **** *p* < 0.0001, ns indicates no significant difference.

**Figure 2 microorganisms-13-01209-f002:**
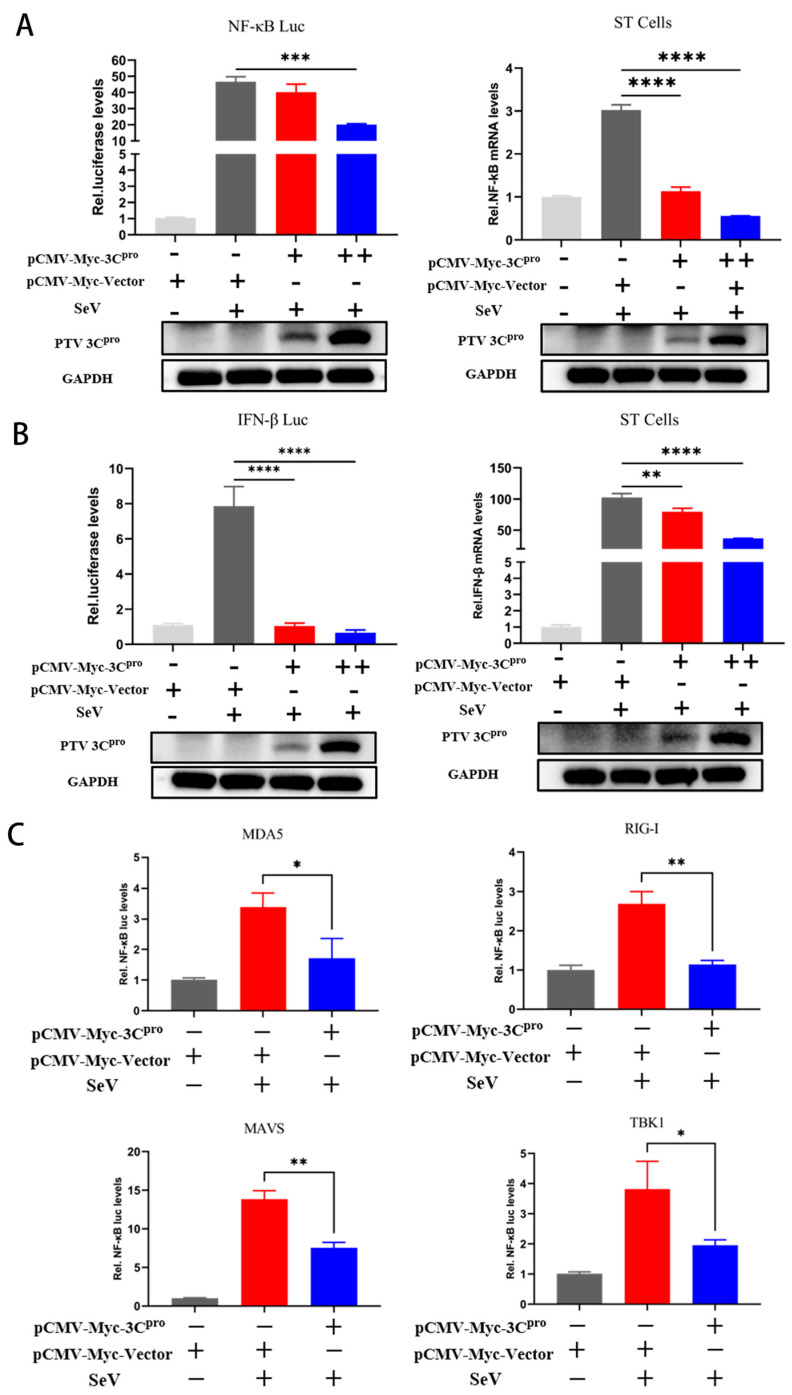
PTV 3C^pro^ inhibits NF-κB and IFN-β mRNA expression and promoter activity. (**A**,**B**) Cotransfection of NF-κB-Luc or IFN-β-Luc (1000 ng) with PRL-TK (50 ng) and 3C^pro^ (200 ng, 500 ng) into 293T cells, followed by 12 h of induction with SeV (MOI = 1) and measurement of firefly luciferase activity, normalized to Renilla luciferase activity. 3C^pro^ (200 ng, 500 ng) was transfected into ST cells, which were then induced with SeV (MOI = 1) for 12 h, and the NF-κB or IFN-β mRNA levels were measured using fluorescence quantitative PCR. (**C**) NF-κB-Luc (1000 ng) with PRL-TK (50 ng) and 3C^pro^ (500 ng) were cotransfected into 293T cells, followed by cotransfection of MDA5, RIG-I, TBK1, and MAVS (700 ng) to induce promoter activity. After 24 h, firefly luciferase activity, normalized to Renilla luciferase activity, was measured. * *p* < 0.05, ** *p* < 0.01, *** *p* < 0.001, and **** *p* < 0.0001.

**Figure 3 microorganisms-13-01209-f003:**
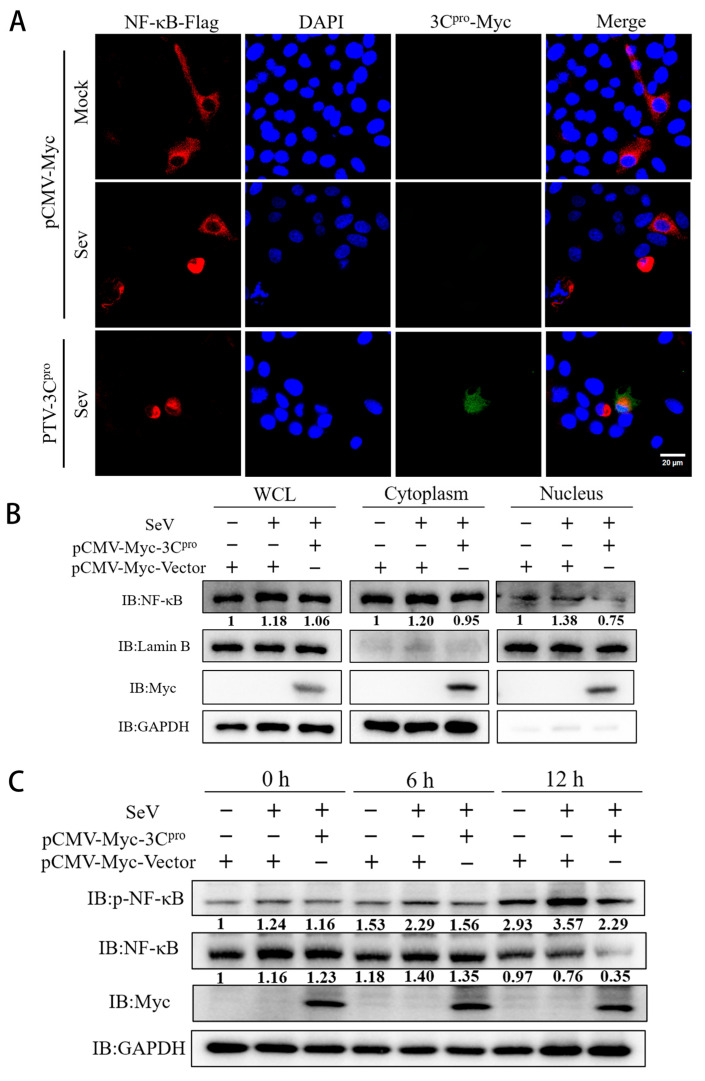
PTV 3C^pro^ inhibits NF-κB phosphorylation and nuclear translocation. (**A**) NF-κB (500 ng) and PTV 3C^pro^ (500 ng) were introduced into ST cells that were infected with SeV after 24 h, and protein fluorescence was observed through an indirect immunofluorescence assay. (**B**) PTV 3C^pro^ (500 ng) was introduced into ST cells infected with SeV after 24 h, and cytoplasmic and nuclear proteins were extracted to measure the levels of NF-κB. GAPDH and Lamin B proteins were used as internal references. Quantitative display is achieved through standardization of the ratio of NF-κB/internal references. (**C**) PTV 3C^pro^ (500 ng) was transfected into ST cells, followed by SeV (MOI = 1) stimulation after 12 h, and the cell lysate was collected at 0 h, 6 h, and 12 h. Western blotting was performed to detect the protein levels of NF-κB and phosphorylated NF-κB. GAPDH protein was used as the internal reference. Quantitative display is achieved through standardization of the ratio of p-NF-κB (NF-κB) to GAPDH.

**Figure 4 microorganisms-13-01209-f004:**
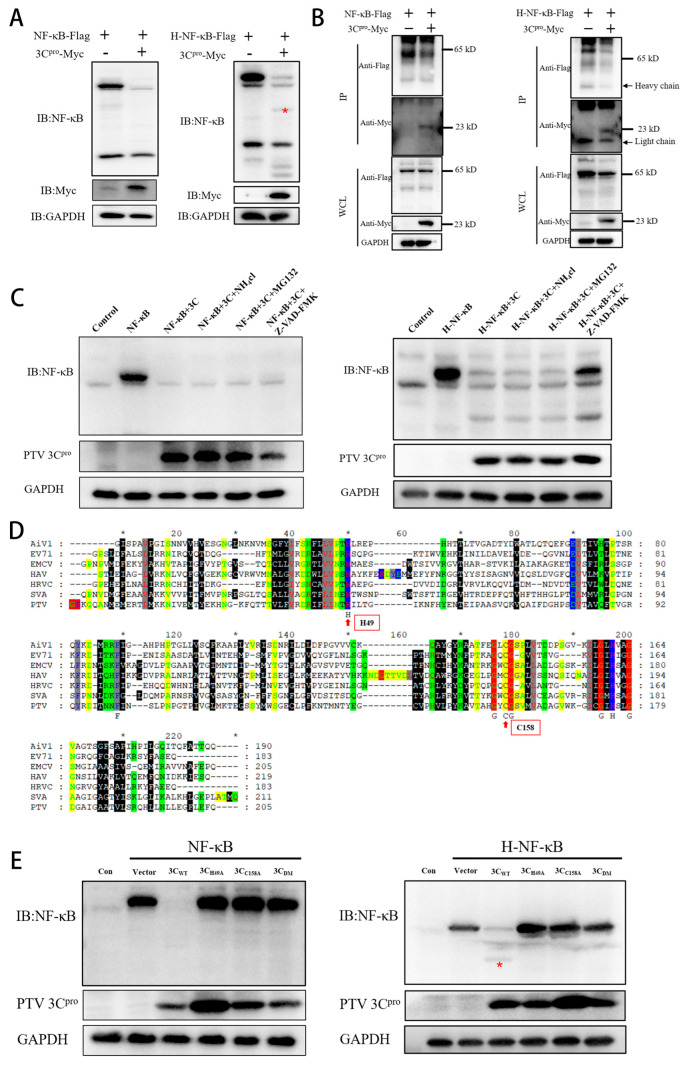
Mechanism of 3C^pro^-mediated degradation of NF-κB. (**A**) PTV 3C^pro^ (1000 ng) was transfected into ST or 293T cells, which were subsequently incubated for 24 h, and NF-κB protein levels were detected using western blotting. The asterisk (*) denotes the band corresponding to H-NF-κB cleaved by PTV 3C^pro^. (**B**) 293T cells were cotransfected with NF-κB (or H-NF-κB) (1000 ng) and the 3C^pro^ plasmid (300 ng), and co-IP and western blotting were subsequently performed to measure the protein levels after 24 h of incubation. (**C**) 293T cells were cotransfected with NF-κB (or H-NF-κB) (1000 ng) and the 3C^pro^ plasmid (1000 ng), followed by 12 h of inhibition treatment and western blotting detection of protein levels. (**D**) Comparative sequence analysis of PTV 3C^pro^ with other members of the *Picornaviridae* family revealed the conserved catalytic site. On the basis of the enzyme activity sites of other viruses, H49 and C158 were predicted to be the enzyme activity sites of PTV 3C^pro^. The asterisks (*) were used to separate consecutive 10-sequence intervals. (**E**) NF-κB (or H-NF-κB) (1000 ng), PTV 3C^pro^, and its mutant (1000 ng) were transfected into 293T cells, which were then cultured for 24 h, after which NF-κB expression was assessed using western blotting. The asterisk (*) denotes the band corresponding to H-NF-κB cleaved by PTV 3C^pro^.

**Figure 5 microorganisms-13-01209-f005:**
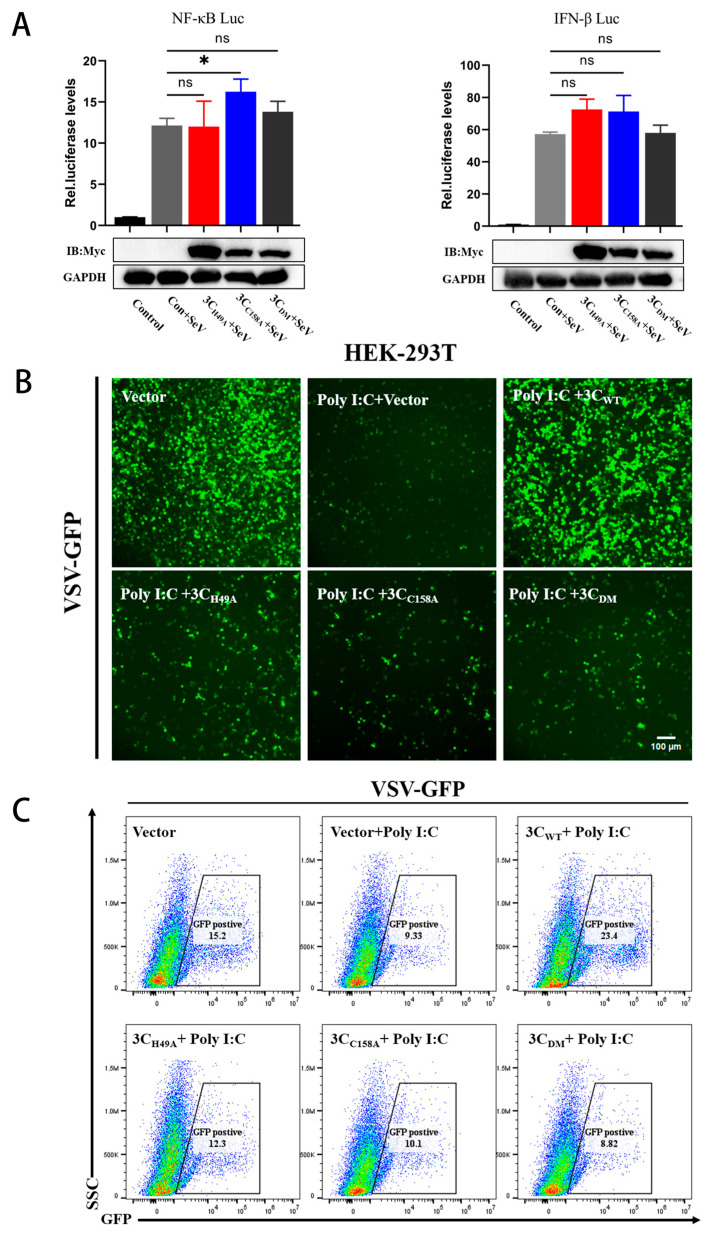
The PTV 3C^pro^ mutant cannot antagonize innate immunity. (**A**) In 293T cells, NF-κB-Luc or IFN-β-Luc (1000 ng) and PRL-TK (50 ng) were cotransfected with 3C_H49A_, 3C_C158A_, or 3C_DM_ (500 ng), followed by SeV induction for 12 h prior to dual-luciferase activity measurement. (**B**,**C**) PTV 3C^pro^ and the variant (1000 ng) were transfected into 293T cells, which were subsequently subjected to VSV infection and poly(I:C) stimulation for 12 h, fluorescence observation, and quantification using fluorescence microscopy and flow cytometry. * *p* < 0.05, ns indicates no significant difference.

**Table 1 microorganisms-13-01209-t001:** Primers for plasmid construction.

Primer	Sequence
NF-κB-F	gatgacgacgataaggaattcATGGACGACCTCTTCCCCC
NF-κB-R	attaagatctgctagctcgagTTAGGAGCTGATCTGACTCAGAAGG
PTV-3C-F	tctgaagaggacttggaattcGGACCAAAGGGACAAGCTAACA
PTV-3C-R	ccgcggccgcggtacctcgagTTGAAATTCCAAAAATCCCTCAA

**Table 2 microorganisms-13-01209-t002:** Primers for SYBR Green I qPCR assays.

Primer	Sequence
Q-GAPDH-F	AGCAACAGGGTGGTGGACCT
Q-GAPDH-R	CTGGGATGGAAACTGGAAGT
Q-NF-κB-F	CCTGAGGCTATAACTCGCTTGG
Q-NF-κB-R	GTCCGCAATGGAGGAGAAGT
Q-IFN-β-F	CATCCTCCAAATCGCTCTCC
Q-IFN-β-R	ACATGCCAAATTGCTGCTCC

**Table 3 microorganisms-13-01209-t003:** Gene-specific mutagenesis primers.

Primer	Sequence
PTV-H49A-F	TCTGATCAATACTgcTATTTTGACAGGTATAAA
PTV-H49A-R	TTTATACCTGTCAAAATAgcAGTATTGATCAGA
PTV-C158A-F	ACGGCTATgcCGGCTCTGTGATGGTTGCGGATGCTGGAG
PTV-C158A-R	CTCCAGCATCCGCAACCATCACAGAGCCGgcATAGCCGT

## Data Availability

The data that support the findings of this study are available from the corresponding author upon reasonable request.

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
