# Peer review of "Porcine Teschovirus 2 3C^pro^ Evades Host Antiviral Innate Immunity by Inhibiting the IFN-β Signaling Pathway"

_microorganisms, 2025, doi:10.3390/microorganisms13061209_

Round 1

Reviewer 1 Report

Comments and Suggestions for Authors

The manuscript “Porcine Teschovirus 2 3Cpro evades host antiviral innate immunity by inhibiting the IFN-β signaling pathway” by Xin-yu Zhang and coauthors have evaluated the mechanisms of action by which PTV antagonizes innate immunity. The authors performed cellular and molecular analyses to show the inhibition of IFN-β and NF-κB phosphorylation, promoting its 3Cpro-mediated degradation. This study is interesting and contributes to our understanding of the immune evasion mechanism caused by PTV infection, particularly PTV-2. However, there are several errors that I consider necessary to be corrected to improve the clarity and presentation of the study.
•    There are several sentences and entire paragraphs identical to a previous study by the authors https://doi.org/10.1016/j.vetmic.2025.110479. This should be corrected, mainly in the methods and discussions.
•    I suggest a complete review of the language, as there are several sections that require improved writing.
Introduction
•    Identify references in brackets here and in other sections of the manuscript as indicated in the journal's author guidelines.
•    Line 59: write enterovirus in detail and then its acronym as in the other cases. 
•    Lines 68-79: These lines primarily contain results. This paragraph should be revised to clearly identify the study's objectives (which preceded the procedures).
Methods
•    Line 88: I suggest writing the text in impersonal verbal form.
•    Section 2.3: Rewrite the paragraph to match the other sections. Methods are usually written in the past tense.
•    Section 2.4: Authors should provide further details on the PCRs performed. If they were designed for the present study, they should detail the reaction conditions. Otherwise, provide the respective references. The same applies to the primers.
•    Section 2.7: Again, authors should detail the PCR conditions used.
Results
•    Figure 2: I suggest adding the quantitative display in a similar way to what was done in Fig 3. In addition, detail these procedures in materials and methods.
•    Lines 262-266: This procedure was previously performed at https://doi.org/10.1016/j.vetmic.2025.110479. For transparency reasons, it is appropriate to mention the reference.
•    Figure 4D: Are the predicted active sites in 3Cpro conserved across all PTV genotypes? I suggest adding an alignment—possibly as supplementary material—of the 3Cs across all PTV genotypes to corroborate this pattern across the species.
Discussion:
•    I suggest adding some lines to define future directions based on the results found. For example, experimental studies in pigs to verify what was shown in vitro. Also, to determine whether the immune inhibition mechanism is directly related to the level of virulence of the PTV strain or genotype.

Comments on the Quality of English Language

•    I suggest a complete review of the language, as there are several sections that require improved writing.

Author Response

The manuscript “Porcine Teschovirus 2 3Cpro evades host antiviral innate immunity by inhibiting the IFN-β signaling pathway” by Xin-yu Zhang and coauthors have evaluated the mechanisms of action by which PTV antagonizes innate immunity. The authors performed cellular and molecular analyses to show the inhibition of IFN-β and NF-κB phosphorylation, promoting its 3Cpro-mediated degradation. This study is interesting and contributes to our understanding of the immune evasion mechanism caused by PTV infection, particularly PTV-2. However, there are several errors that I consider necessary to be corrected to improve the clarity and presentation of the study.

Response: We extend our sincere gratitude for you to provide constructive feedback to enhance its quality. All suggestions have been thoroughly incorporated into the revised manuscript to strengthen methodological clarity, refine data interpretation, and improve overall scientific rigor. We remain deeply appreciative of this opportunity to refine our research presentation and trust that the revisions fully address the valuable insights raised during the review process.

1: There are several sentences and entire paragraphs identical to a previous study by the authors https://doi.org/10.1016/j.vetmic.2025.110479. This should be corrected, mainly in the methods and discussions.

Response: We sincerely appreciate for your insightful comment. We have thoroughly revised the manuscript to address overlapping content with our prior work , particularly in the Methods and Discussion sections, by rephrasing or removing redundant text. We confirm that all remaining content in this manuscript is original and not directly reused from prior publications.

2: I suggest a complete review of the language, as there are several sections that require improved writing.

Response: Thank you for your valuable feedback. We have carefully revised the manuscript to enhance clarity, grammar, and overall readability, particularly in the sections highlighted. In addition, professional editing services were engaged to ensure adherence to academic writing standards and enhance the readability and precision of the text.

Introduction

3: Identify references in brackets here and in other sections of the manuscript as indicated in the journal's author guidelines.

Response: We sincerely appreciate your valuable reminders and suggestions. We have meticulously reviewed and revised the manuscript’s references to ensure strict adherence to the journal’s formatting requirements. Additionally, We resolved citation formatting discrepancies caused by PDF export settings.

4: Line 59: write enterovirus in detail and then its acronym as in the other cases.

Response: Thank you for your careful review and feedback. We have updated the text to spell out "enterovirus 71" in full at its first mention.

5: Lines 68-79: These lines primarily contain results. This paragraph should be revised to clearly identify the study's objectives (which preceded the procedures).

Response: Thank you for your constructive comments. We have revised the paragraph to explicitly clarify the research objectives and purpose of the study, as recommended.

Although significant progress has been made in understanding the pathogenesis and immune evasion strategies of PTV, further investigation is still needed. Given the crucial role of picornaviral 3Cpro in viral replication and host antiviral pathway suppression, elucidating its functional mechanisms in PTV immune evasion holds important theoretical significance and research value. Therefore, this study aimed to elucidate the functional significance of PTV 3Cpro in interfering with key antiviral pathways. By exploring these mechanisms, we hope to provide new insights into PTV pathogenesis and contribute to the development of targeted interventions against this important swine pathogen. (Page 2, Line 80-88)

Methods

6: Line 88: I suggest writing the text in impersonal verbal form.

Response: We appreciate your suggestion and apologize for any potential confusion. The text has been revised to comply with academic writing conventions. (Page 3, Line 111)

7: Section 2.3: Rewrite the paragraph to match the other sections. Methods are usually written in the past tense.

Response: Thank you for your feedback. We have revised the verb tenses in the specified paragraph to fully comply with academic writing standards.

293T or ST cells were seeded in 12-well plates and cotransfected with reporter plasmids encoding NF-κB-Luc or IFN-β-Luc, along with TK-Renilla and other necessary expression plasmids, using ExFect Transfection Reagent (Vazyme, T101). An empty vector served as a negative control. A TransDetect® Double-Luciferase Reporter Assay Kit (TransGen, Lot# R30906) was used for dual luciferase detection. The luciferase reac-tion reagent and luciferase reaction reagent II were sequentially added according to the manufacturer’s instructions. The luminescence values of Luc and TK were measured using a microplate reader (BioTek, Synergy H1), and the luciferase activity was normalized to firefly luciferase and Renilla luciferase values. All data from the reporter gene assays were derived from three independent experiments. (Page 3, Line 124-133)

8: Section 2.4: Authors should provide further details on the PCRs performed. If they were designed for the present study, they should detail the reaction conditions. Otherwise, provide the respective references. The same applies to the primers.

Response: Thank you for your valuable suggestion. We have added the relevant PCR protocols in the corresponding section of the manuscript.

According to the instructions included with the high-fidelity PCR Master Mix (Vazyme, P525), the PCR protocol was set as follows: initial heat activation at 95°C for 3 min, followed by 35 cycles of amplification (each cycle consisting of 95°C denaturation for 15 s, 58°C annealing for 15 s, and 72°C extension for 1 min), with a final extension at 72°C for 5 min. (Page 4, Line 151-155)

9: Section 2.7: Again, authors should detail the PCR conditions used.

Response: Thank you for your careful review and suggestions. As per previous recommendations, we have explicitly included the complete PCR protocol in Section 2.3 (initial description). To maintain conciseness, we referenced this protocol in Section 2.7, where the annealing temperature was adjusted to 65°C.

The PCR amplification was performed using the same protocol as described in section 2.3, except with an annealing temperature of 65°C. (Page 6, Line 221-223)

10: Figure 2: I suggest adding the quantitative display in a similar way to what was done in Fig 3. In addition, detail these procedures in materials and methods.

Response: We sincerely thank you for your valuable comments and apologize for any confusion. We have added detailed descriptions of the relevant experimental procedures in the "Materials and Methods" section to facilitate a clearer understanding of the data analysis. Figure 2 uses bar charts to clearly illustrate the relative expression trends of dual fluorescence and mRNA levels. To maintain the clarity of the figure, quantitative analysis has not been included. Should further modifications be required, we are open to incorporating the suggested revisions.

Grayscale analysis was performed using ImageJ software: images were converted to 8-bit grayscale mode, and background noise was removed using default parameters. After inverting the image, the area, mean gray value, and integrated density were measured. Data were normalized for subsequent statistical analy-sis. (Page 6, Line 209-212)

11: Lines 262-266: This procedure was previously performed at https://doi.org/10.1016/j.vetmic.2025.110479. For transparency reasons, it is appropriate to mention the reference.

Response: Thank you for your careful review and valuable suggestion. The reference has now been properly cited in the revised manuscript. Initially, this work was not cited as it was under peer review at the time of our submission. Now that it has been published, we have incorporated it into the manuscript accordingly.

12: Figure 4D: Are the predicted active sites in 3Cpro conserved across all PTV genotypes? I suggest adding an alignment—possibly as supplementary material—of the 3Cs across all PTV genotypes to corroborate this pattern across the species.

Response: Thank you for this constructive suggestion. A multiple sequence alignment of 3C proteases across all analyzed PTV genotypes has been included as Supplementary Figure S1. The alignment confirms the complete conservation of the catalytic residues H49 and C158 across all PTV genotypes.

Discussion

13: I suggest adding some lines to define future directions based on the results found. For example, experimental studies in pigs to verify what was shown in vitro. Also, to determine whether the immune inhibition mechanism is directly related to the level of virulence of the PTV strain or genotype.

Response: Thank you for your valuable suggestions. We have incorporated future research directions into the revised manuscript. Specifically, we have expanded the Discussion section to elaborate on the need for in vivo validation of the in vitro findings and to explore the potential association between the immune suppression mechanism and the virulence of PTV strains/genotypes.

Future research should validate these findings in in vivo porcine models to assess how 3Cpro-mediated immune suppression impacts viral pathogenesis and tissue tropism. Comparative analyses of PTV-1 and PTV-2 strains, including the HeNZ1 variant, are critical to determine whether genotype-specific 3Cpro activity correlates with virulence and immune evasion efficiency. Reverse genetic approaches should generate 3Cpro-mutant PTV viruses to evaluate their impact on immune evasion; however, prior to assessing immune evasion, the effect of 3Cpro inactivation on viral replication must be established. (Page 17, Line 527-534)

Comments on the Quality of English Language

14: I suggest a complete review of the language, as there are several sections that require improved writing.

Response: We sincerely appreciate your constructive feedback on improving the manuscript's language quality. We have performed a comprehensive language revision of the manuscript with the assistance of a third-party professional editing service, addressing grammatical errors, syntactical ambiguities, and terminology inconsistencies section-by-section. We appreciate your guidance and are committed to ensuring the language meets the highest academic standards.

Reviewer 2 Report

Comments and Suggestions for Authors

This article provides information on how porcine teschovirus 2 3Cpro evades host antiviral innate immunity. It is in general appropriately organized, carried out and written, however there are some points that should be corrected or clarified. Please check comments and corrections in the attached file.

Author Response

This article provides information on how porcine teschovirus 2 3Cpro evades host antiviral innate immunity. It is in general appropriately organized, carried out and written, however there are some points that should be corrected or clarified. Please check comments and corrections in the attached file.

Response: Thank you for your constructive feedback and thorough evaluation of our manuscript. We have carefully addressed all the points raised in your comments and have made comprehensive revisions to improve clarity and accuracy. The manuscript has been updated in full accordance with your suggestions to ensure the highest quality of presentation.

1: Line 33: Please check reference style of the journal.

Response: Thank you for your thorough review. We have verified and revised the citation format of the referenced literature.

2: Line 64: Please explain abbreviations when they are initially used in text

Response: Thank you for your valuable suggestions. We have added the full name of MDA5, Melanoma differentiation-associated gene 5.

3: Line 68-79: This part is not necessary here. You can use it at the end as "Conclusions". Here, please add the aim of your study

Response: Thank you for your valuable advices. Based on your suggestions, I have removed the original conclusion section and supplemented the research objectives in its place to more clearly articulate the purpose and significance of this study.

Although significant progress has been made in understanding the pathogenesis and immune evasion strategies of PTV, further investigation is still needed. Given the crucial role of picornaviral 3Cpro in viral replication and host antiviral pathway suppression, elucidating its functional mechanisms in PTV immune evasion holds important theoretical significance and research value. Therefore, this study aimed to elucidate the functional significance of PTV 3Cpro in interfering with key antiviral pathways. By exploring these mechanisms, we hope to provide new insights into PTV pathogenesis and contribute to the development of targeted interventions against this important swine pathogen. (Page 2, Line 80-88)

4: Line 87 : Which laboratory?

Response: Thank you for your reminder. The name of the laboratory has been updated to Wenzhou Key Laboratory for Virology and Immunology.

5: Line 88: Please provide the initials of the manufacturer (that possibly is a co-author of the study)

Response: Thank you for your reminder. The manufacturer's name has been updated.

6: Line 98-106: Please convert from imperative to indicative mood

Response: Thank you for your suggestion. We have revised the text to convert all imperative statements to the indicative mood while preserving the original meaning and experimental details.

293T or ST cells were seeded in 12-well plates and cotransfected with reporter plasmids encoding NF-κB-Luc or IFN-β-Luc, along with TK-Renilla and other necessary expression plasmids, using ExFect Transfection Reagent (Vazyme, T101). An empty vector served as a negative control. A TransDetect® Double-Luciferase Reporter Assay Kit (TransGen, Lot# R30906) was used for dual luciferase detection. The luciferase reac-tion reagent and luciferase reaction reagent II were sequentially added according to the manufacturer’s instructions. The luminescence values of Luc and TK were measured using a microplate reader (BioTek, Synergy H1), and the luciferase activity was normalized to firefly luciferase and Renilla luciferase values. All data from the reporter gene assays were derived from three independent experiments. (Page 3, Line 124-133)

7: Line 223: Please explain what each bar represents

Response: We sincerely thank you for your valuable comments and apologize for any confusion. We have revised and updated the figure caption to more clearly indicate the meaning of each bar for better clarity.

(A and B) Cotransfection of NF-κB-Luc or IFN-β-Luc (1000 ng) with PRL-TK (50 ng) and 3C (200 ng, 500 ng) into 293T cells, followed by 12 hours of induction with SeV (MOI=1) and measurement of firefly luciferase activity, normalized to Renilla luciferase activity. 3C (200 ng, 500 ng) was transfected into ST cells, which were then induced with SeV (MOI=1) for 12 h, and the NF-κB or IFN-β mRNA levels were detected through fluorescence quantitative PCR. (C) Cotransfection of NF-κB-Luc (1000 ng) with PRL-TK (50 ng) and 3C (500 ng) into 293T cells, followed by cotransfection of MDA5, RIG-I, TBK1, and MAVS (700 ng) to induce promoter activity, and after 24 h, the measurement of firefly luciferase activity, normalized to Renilla luciferase activity, was performed. (Page 10, Line 311-320)

8: Line 326: Reference?

Response: We sincerely thank you for your reminder. We have added the appropriate references at the indicated position in the manuscript. (Page 16, Line 466)

9: Line 370: Please add a section "Conclusions"

Response: We sincerely thank you for your valuable comments and apologize for any confusion. A Conclusion section has been added to the manuscript.

In summary, our findings demonstrate that PTV 3Cpro plays a critical role in evading the host innate immune response by binding with NF-κB, inhibiting its phosphorylation and nuclear translocation. Notably, we show that PTV 3Cpro-induced degradation of NF-κB is species-specific. The results of mutagenesis studies further confirmed that the protease activity of 3Cpro is essential for these immunosuppressive functions. These re-sults provide novel insights into the immune evasion strategies of PTV and highlight the importance of 3Cpro as a potential target for antiviral intervention. (Page 17-18, Line 535-541)

10: Line 50: “A great research question is how PTV” can replace “how PTV can”.

Response: We sincerely thank for the suggestion. We have revised the sentence to:

"A great research question is how PTV can evade innate immunity and spread through replication." (Page 2, Line 57)

11: Line 15: The word 'elucidated' replaces 'characterized'.

Response: Thank you very much for this helpful suggestion. We have replaced “characterized” with “elucidated” as recommended.

Round 2

Reviewer 1 Report

Comments and Suggestions for Authors

The authors have satisfactorily responded to all the observations made.